# Feature Map Transform Coding for Energy-Efficient CNN Inference

## Abstract

Convolutional neural networks (CNNs) achieve state-of-the-art accuracy in a variety of tasks in computer vision and beyond. One of the major obstacles hindering the ubiquitous use of CNNs for inference on low-power edge devices is their high computational complexity and memory bandwidth requirements. The latter often dominates the energy footprint on modern hardware. In this paper, we introduce a lossy transform coding approach, inspired by image and video compression, designed to reduce the memory bandwidth due to the storage of intermediate activation calculation results. Our method does not require fine-tuning the network weights and halves the data transfer volumes to the main memory by compressing feature maps, which are highly correlated, with variable length coding. Our method outperform previous approach in term of the number of bits per value with minor accuracy degradation on ResNet-34 and MobileNetV2. We analyze the performance of our approach on a variety of CNN architectures and demonstrate that FPGA implementation of ResNet-18 with our approach results in a reduction of around 40% in the memory energy footprint, compared to quantized network, with negligible impact on accuracy. When allowing accuracy degradation of up to 2%, the reduction of 60% is achieved. A reference implementation accompanies the paper.

## 1 Introduction

Deep neural networks have established themselves as the first-choice tool for a wide range of applications. Neural networks have shown phenomenal results in a variety of tasks in a broad range of fields such as computer vision, computational imaging, and image and language processing. Despite deep neural models impressive performance, the computation and computational requirements are substantial for both training and inference phases. So far, this fact has been a major obstacle for the deployment of deep neural models in applications constrained by memory, computational, and energy resources, such as those running on embedded systems.

To alleviate the energy cost, custom hardware for neural network inference, including FPGAs and ASICs, is actively being developed in recent years. In addition to providing better energy efficiency per arithmetic operation, custom hardware offers more flexibility in various strategies to reduce the computational and storage complexity of the model inference, for example by means of quantization (Baskin et al., 2018; Hubara et al., 2018; Jacob et al., 2018) and pruning (Han et al., 2016; Louizos et al., 2018; Theis et al., 2018). In particular, quantization to very low precision is especially efficient on custom hardware where arbitrary precision arithmetic operations require proportional resources. To prevent accuracy degradation, many approaches have employed training the model with quantization constraints or modifying the network structure.

A recent study (Yang et al., 2017) has shown that almost 70% of the energy footprint on such hardware is due to data movement to and from the off-chip memory. Amounts of data typically need to be transferred to and from the RAM and back during the forward pass through each layer, since the local memory is too small to store all the feature maps. By reducing the number of bits representing these data, existing quantization techniques reduce the memory bandwidth considerably. However, to the best of our knowledge, none of these methods exploit the high amount of interdependence between the feature maps and spatial locations of the compute activations.

**Contributions.** In this paper, we propose a novel scheme based on transform-domain quantization of the neural network activations followed by lossless variable length coding. We demonstrate that this approach reduces memory bandwidth by 40% when applied in the post-training regime (i.e., without fine-tuning) with small computational overhead and no accuracy degradation. Relaxing the accuracy requirements increases bandwidth savings to 60%. Moreover, we outperform previous methods in term of number bit per value with minor accuracy degradation. A detailed evaluation of various ingredients and parameters of the proposed method is presented. We also demonstrate a reference hardware implementation that confirms a reduction in memory energy consumption during inference.

## 2 Related work

**Quantization.** Low-precision representation of the weights and activations is a common means of reducing computational complexity. On appropriate hardware, this typically results in the reduction of the energy footprint as well. It has been demonstrated that in standard architectures quantization down to 16 (Gupta et al., 2015) or 8 bits (Jacob et al., 2018; Lee et al., 2018; Yang et al., 2019) per parameter is practically harmless. However, further reduction of bitwidth requires non-trivial techniques (Mishra et al., 2018; Zhang et al., 2018), often with adverse impact on training complexity. Lately, the quantization of weights and activations of neural networks to 2 bits or even 1 (Rastegari et al., 2016; Hubara et al., 2018) has attracted the attention of many researchers. While the performance of binary (i.e., 1-bit) neural networks still lags behind their full-precision counterparts (Ding et al., 2019; Peng & Chen, 2019), existing quantization methods allow 2-4 bit quantization with a negligible impact on accuracy (Choi et al., 2018b;a; Dong et al., 2019).

Quantizing the neural network typically requires introducing the quantizer model at the training stage. However, in many applications the network is already trained in full precision, and there is no access to the training set to configure the quantizer. In such a *post-training* regime, most quantization methods employ *statistical clamping*, i.e., the choice of quantization bounds based on the statistics acquired in a small test set. Migacz (2017) proposed using a small calibration set to gather activation statistics and then randomly searching for a quantized distribution that minimizes the Kullback-Leibler divergence to the continuous one. Gong et al. (2018), on the other hand, used the $L_\infty$ norm of the tensor as a threshold. Lee et al. (2018) employed channel-wise quantization and constructed a dataset of parametric probability densities with their respective quantized versions; a simple classifier was trained to select the best fitting density. Banner et al. (2018) derived an approximation of the optimal threshold under the assumption of Laplacian or Gaussian distribution of the weights, which achieved single-percentage accuracy reduction for 8-bit weights and 4-bit activations. Meller et al. (2019) showed that the equalization of channels and removal of outliers improved quantization quality. Choukroun et al. (2019) used one-dimensional line-search to evaluate an optimal quantization threshold, demonstrating state-of-the-art results for 4-bit weight and activation quantization.

**Influence of memory access on energy consumption.** Yang et al. (2017) studied the breakdown of energy consumption in CNN inference. For example, in GoogLeNet (Szegedy et al., 2015) arithmetic operations consume only 10% of the total energy, while feature map transfers to and from an external RAM amount to about 68% of the energy footprint. However, due to the complexity of real memory systems, not every method that decreases the sheer memory bandwidth will necessarily yield significant improvement in power consumption. In particular, it depends on computational part optimization: while memory performance is mainly defined by the external memory chip, better optimization of computations will lead to higher relative energy consumption of the memory. For example, while Ansari & Ogunfunmi (2018) reported a 70% bandwidth reduction, the dynamic power consumption decreased by a mere 2%.

Xiao et al. (2017) proposed fusing convolutional layers to reduce the transfer of feature maps. In an extreme case, all layers are fused into a single group. A similar approach was adopted by Xing et al. (2019), who demonstrated a hardware design that does not write any intermediate results into the off-chip memory. This approach achieved approximately 15% runtime improvement for ResNet and state-of-the-art throughput. However, the authors did not compare the energy footprint of the design with the baseline. Morcel et al. (2019) demonstrated that using on-chip cache cuts down the memory bandwidth and thus reduces power consumption by an order of magnitude. In addition, Jouppi et al.

(2017) noted that not only the power consumption but also the speed of DNN accelerators is memory-rather than compute-bound. This is confirmed by Wang et al. (2019), who also demonstrated that increasing computation throughput without increasing memory bandwidths barely affects latency.

**Network compression.** Lossless coding and, in particular, variable length coding (VLC), is a way to reduce the memory footprint without compromising performance. In particular, Han et al. (2016) proposed using Huffman coding to compress network weights, alongside quantization and pruning. Wijayanto et al. (2019) proposed using the more computationally-demanding algorithm DEFLATE (LZ77 + Huffman) to further improve compression rates. Chandra (2018) used Huffman coding to compress feature maps and thus reduce memory bandwidth. Gudovskiy et al. (2018) proposed passing only the lower-dimensional feature maps to the memory to reduce the bandwidth. Cavigelli & Benini (2019) proposed using RLE-based algorithm to compress sparse network activations.

# 3 Transform-Domain Compression

In what follows, we briefly review the basics of lossy transform coding. For a detailed overview of the subject, we refer the reader to Goyal (2001). Let $\mathbf{x} = (x_1, \ldots, x_n)$ represent the values of the activations of a NN layer in a block of size $n = W \times H \times C$ spanning, respectively, the horizontal and the vertical dimensions and the feature channels. Prior to being sent to memory, the activations, $\mathbf{x}$, are encoded by first undergoing an affine transform, $\mathbf{y} = \mathcal{T}\mathbf{x} = \mathbf{T}(\mathbf{x} - \boldsymbol{\mu})$; the transform coefficients are quantized by a scalar quantizer, $\mathcal{Q}_\Delta$, whose strength is controlled by the step size $\Delta$, and subsequently coded by a lossless variable length coder (VLC). We refer to the length in bits of the resulting code, normalized by $n$ as to the *average rate*, $R_\Delta$. To decode the activation vector, a variable length decoder (VLD) is applied first, followed by the inverse quantizer and the inverse transform. The resulting decoded activation, $\hat{\mathbf{x}} = \mathcal{T}^{-1}\mathcal{Q}_\Delta^{-1}(\mathcal{Q}_\Delta(\mathcal{T}\mathbf{x}))$, typically differs from $\mathbf{x}$; the discrepancy is quantified by a *distortion*, $D_\Delta$. The functional relation between the rate and the distortion is controlled by the quantization strength, $\Delta$, and is called *rate-distortion curve*.

Classical rate-distortion analysis in information theory assumes the MSE distortion, $D = \frac{1}{n}\|\mathbf{x} - \hat{\mathbf{x}}\|_2^2$. While in our case the measure of distortion is the impact on the task-specific loss, we adopt the Euclidean distortion for two reasons: firstly, it is well-studied and leads to simple expressions for the quantizer; and, secondly, computing loss requires access to the training data, which are unavailable in the post-training regime. The derivation of an optimal rate is provided in Appendix D.

A crucial observation justifying transform coding is the fact that significant statistical dependence usually exists between the $x_i$ (Cogswell et al., 2016). We model this fact by asserting that the activations are jointly Gaussian, $\mathbf{x} \sim \mathcal{N}(\boldsymbol{\mu}, \boldsymbol{\Sigma})$, with the covariance matrix $\boldsymbol{\Sigma}$, whose diagonal elements are denoted by $\sigma_i^2$. Statistical dependence corresponds to non-zero off-diagonal elements in $\boldsymbol{\Sigma}$. The affinely transformed $\mathbf{y} = \mathbf{T}(\mathbf{x} - \boldsymbol{\mu})$ is also Gaussian with the covariance matrix $\boldsymbol{\Sigma}' = \mathbf{T}^{\mathrm{T}}\boldsymbol{\Sigma}\mathbf{T}$. The distortion (D.2) is minimized over orthonormal matrices by $\mathbf{T} = \boldsymbol{\Sigma}^{-1/2}$ diagonalizing the covariance (Goyal, 2001). The latter is usually referred to as the Karhunen-Loeve transform (KLT) or principal component analysis (PCA). The corresponding minimum distortion is $D^*(R) = \frac{\pi e}{6}\det(\boldsymbol{\Sigma})^{1/n}2^{-2R}$. Since the covariance matrix is symmetric, $\mathbf{T}$ is orthonormal, implying $\mathbf{T}^{-1} = \mathbf{T}^{\mathrm{T}}$.

In Fig. 1, a visualization of 2D vector quantization is shown. For correlated channels (Fig. 1a), many 2D quantization bins are not used since they contain no values. Linear transformation (Fig. 1b) provides improved quantization error for correlated channels by getting rid of those empty bins.

## 3.1 Implementation

In what follows, we describe an implementation of the transform coding scheme at the level of individual CNN layers. The convolutional layer depicted in Fig. 2 comprises a bank of convolutions (denoted by $*$ in the Figure) followed by batch normalization (BN) that is computed on an incoming input stream. The output of BN is a 3D tensor that is subdivided into 3D blocks to which the transform coding is applied. Each such block is sent to an encoder, where it undergoes PCA, scalar quantization and VLC. The bit stream at the VLC output has a lower rate than the raw input and is accumulated

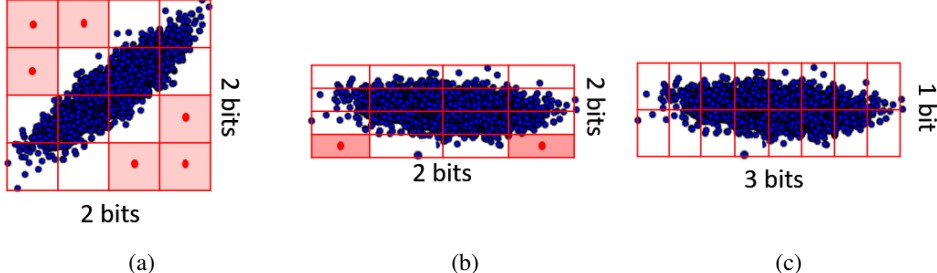

(a)          (b)          (c)

Figure 1: Vector quantization in 2D case. **(a)** A pair of correlated channels on a scatter plot. All values in a cell are mapped to the center of the cell; hence, small cells induce small quantization noise. Several bins are empty (red cells); **(b)** Decorrelation improves utilization since the cells are smaller now; **(c)** Forcing equal bin size along all dimensions further improves utilization. Instead of restricting both channel dynamic range to be divided into same number of bins, we use uniform bin size along all dimensions. VLC allows to further compress the representation since the channels with smaller dynamic range have are mapped mostly to a few most probable bins.

in the external memory. Once all the output of the layer has been stored in the memory, it can be streamed as the input to the following layer. For that purpose, the inverse process is performed by the decoder: a VLD produces the quantized levels that are scaled back to the transform domain, and an inverse PCA is applied to reconstruct each of the activation blocks. The layer non-linearity is then applied, and the activations are used as an input to the following layer. While the location of the nonlinearity could also precede the encoder, our experiments show that the described scheme performs better.

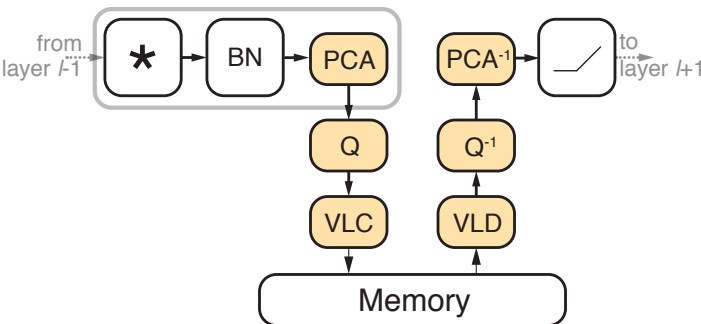

Figure 2: High-level flow diagram of the encoder-decoder chain. PCA and BN are folded into the convolution weights (denoted by $*$), resulting in a single convolution (boxed in grey).

**Linear transform.** We have explored different sizes of blocks for the PCA transform and found $1 \times 1 \times C$ to be optimal (the ablation study is shown in Appendix B). Due to the choice of $1 \times 1 \times C$ blocks, the PCA transform essentially becomes a $1 \times 1$ tensor convolution kernel (Fig. E.1 in the Appendix shows an example of its application to an image). This allows further optimization: as depicted in Fig. 2, the convolution bank of the layer, BN and PCA can be folded (Jacob et al., 2018) into a single operation, offering also an advantage in the arithmetic complexity.

The PCA matrix is pre-computed, as its computation requires the expensive eigendecomposition of the covariance matrix. The covariance matrix is estimated on a small batch of (unlabeled) training or test data and can be updated online. Estimation of the covariance matrix for all layers at once is problematic since quantizing activations in the $l$-th layer alters the input to the $l + 1$-st layer, resulting in a biased estimation of the covariance matrix in the $l + 1$-st layer. To avoid this, we calculate the covariance matrix layer by layer, gradually applying the quantization. The PCA matrix is calculated after quantization of the weights is performed, and is itself quantized to $8$ bits.

**Quantization.** For transformed feature maps we use a uniform quantization, where the dynamic range is determined according to the channel with the highest variance. Since all channels have an equal quantization step, entropy of the low-variance channels is significantly reduced.

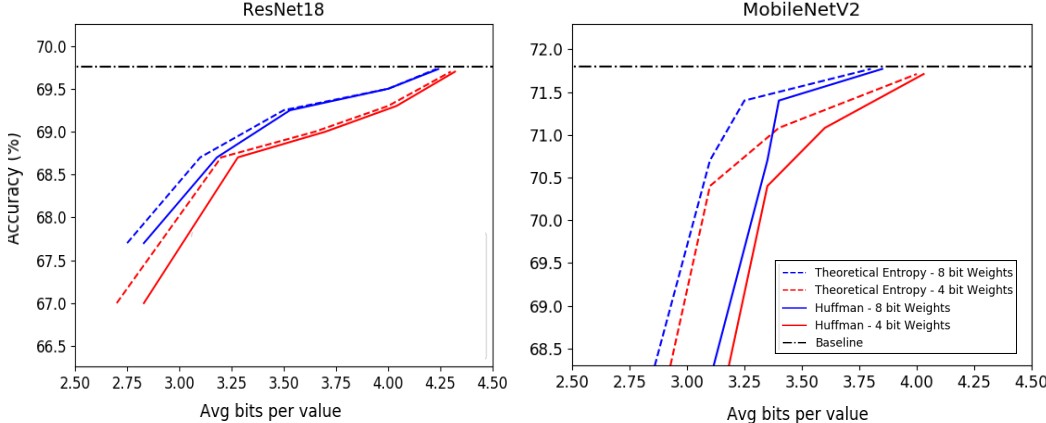

Figure 3: Rate-distortion curves for ResNet-18 and MobileNetV2 architectures with 8- (blue) and 4-bit (red) weight quantization. Distortion is evaluated in terms of top-1 accuracy on ImageNet. Dashed lines represent rates obtained by Huffman VLC, while solid lines represent theoretical rates (entropy). More architectures can be found in Fig. A.1 in the Appendix.

**Variable length coding.** The theoretical rate associated with a discrete random variable, $Y$ (the output of the quantizer), is given by its entropy $H(X) = -\mathbb{E}\log_2 X = -\sum_i p(x_i)\log_2 p(x_i)$. This quantity constitutes the lower bound on the amount of information required for lossless compression of $Y$. We use Huffman codes, which are a practical variable length coding method (Szpankowski, 2000), achieving the rates bounded by $H(X) \leq R \leq H(X) + 1$ (see Fig. 3 for a comparison of the theoretical rates to the ones attained by Huffman codes). Fig. F.1 in the Appendix shows an example of Huffman trees constructed directly on the activations and their PCA coefficients.

$1 \times 1$ **and grouped convolutions.** While for regular $3 \times 3$ convolutions the computational overhead is small, there are two useful cases in which this is not true: $1 \times 1$ and grouped convolutions. For $1 \times 1$ convolutions the overhead is higher: the transformation requires as much computation as the convolution itself. Nevertheless, it can still be feasible in the case of energy-efficient computations. In the case of grouped convolutions, it is impossible to fold the transformation inside the convolution. However, in the common case when the grouped convolution is followed by a regular one, we can change the order of operations: we perform BN, activation and transformation before writing to the memory. This way, the inverse transformation can be folded inside the following convolution.

# 4   Experimental Results

We evaluate the proposed framework on common CNN architectures that have achieved high performance on the ImageNet benchmark. The inference contains 2 stages: a calibration stage, on which the linear transformation is learned based on a single batch of data, and the test stage.

**Full model performance.** We evaluted our method on different CNN architectures: ResNet-18, 50, and 101 (He et al., 2016); MobileNetV2 (Sandler et al., 2018); and InceptionV3 (Szegedy et al., 2016). Specifically, MobileNetV2 is known to be unfriendly to activation quantization (Sheng et al., 2018). Performance was evaluated on ImageNet dataset (Russakovsky et al., 2015) on which the networks were pre-trained. The proposed method was applied to the outputs of all convolutional layers, while the weights were quantized to either 4 or 8 bits (two distinct configurations) using the method proposed by Banner et al. (2018). Rates are reported both in terms of the entropy value and the average length of the feature maps compressed using Huffman VLC in Fig. 3 and Fig. A.1 in the Appendix. We observed that higher compression is achieved for covariance matrices with fast decaying eigenvalues describing low-dimensional data. A full analysis can be found in Appendix G.

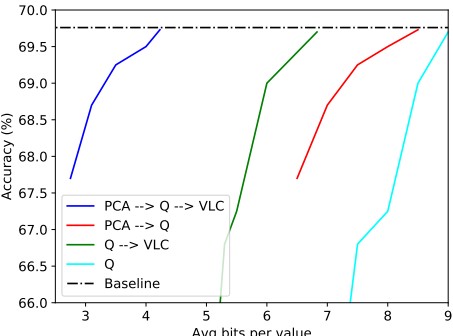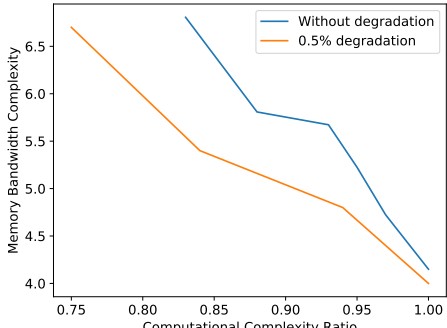

Figure 4: Ablation study of the proposed encoder on ResNet-18. Left: rate-distortion curve with different encoder configurations. Theoretical rates are reported; top-1 accuracy is used as the distortion measure. Right: theoretical memory rate in bits per value achieved for different levels of PCA truncation for baseline and 0.5% lower than baseline top-1 accuracy.

Table 1: Comparison with EBPC (Cavigelli & Benini, 2019). While EBPC does not affect performance of the network, our method allows better compression by exploiting rate-distortion tradeoff.

| Architecture | Method | Activations (avg. number of bits per value) | Accuracy |
|---|---|---|---|
| | EBPC | 3.33 | 73.3% |
| ResNet-34 | Our method | 3.9 | 72.9% |
| | Our method | **3.11** | 72.1% |
| | EBPC | 3.64 | 71.7% |
| MobileNetV2 | Our method | 3.8 | 71.6% |
| | Our method | **3.25** | 71.4% |

**Comparison to other methods.** We compare the proposed method with other post-training quantization methods: ACIQ (Banner et al., 2018), GEMMLOWP (Jacob & Warden, 2017), and KLD (Migacz, 2017). Note that our method can be applied on top of any of them to further reduce the memory bandwidth. For each method, we varied the bitwidth and chose the smallest one that attained top-1 accuracy within 0.1% from the baseline and measured the entropy of the activations. Our method reduces, in average, 36% of the memory bandwidth relatively to the best competing methods; the full comparison can be found in Table A.1 in the Appendix.

As for other memory bandwidth reduction methods, our method shows better performance than Cavigelli & Benini (2019) at the expense of performance degradation (Table 1). The performance difference is smaller in MobileNetV2, since mobile architectures tend to be less sparse (Park et al., 2018), making RLE less efficient. While the method proposed by Gudovskiy et al. (2018) requires fine-tuning, our method, although introducing computational overhead, can be applied to any network without such limitations. In addition, similarly to (Gudovskiy et al., 2018) it is possible to compress only part of the layers in which the activation size is most significant.

**Ablation study.** An ablation study was performed using ResNet-18 to study the effect of different ingredients of the proposed encoder-decoder chain. The following settings were compared: direct quantization of the activations; quantization of PCA coefficients; direct quantization followed by VLC; and full encoder chain comprising PCA, quantization, and VLC. The resulting rate-distortion curves are compared in Fig. 4 (left). Our results confirm the previous result of Chandra (2018), suggesting that VLC applied to quantized activations can significantly reduce memory bandwidth. They further show that a combination with PCA makes the improvement dramatically bigger. In addition, we analyze the effect of truncating the least significant principal components, which reduces the computational overhead of PCA. Fig. 4 (right) shows the tradeoff between the computational and memory complexities, with baseline accuracy and $0.5\%$ below the baseline.

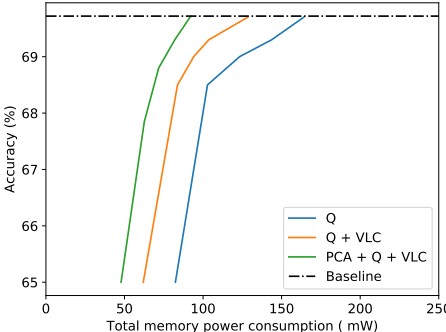

Figure 5: Top-1 accuracy on ResNet18 ImageNet vs. power consumption of our hardware implementation. Each point represents a different quantization rate.

# 5 Hardware Implementation

In order to verify the practical impact of the proposed approach of reducing feature map entropy to save total energy consumption, we implemented the basic building blocks of a pre-trained ResNet-18, with weights and activation quantized to 8 bit, on Intel Stratix-10 FPGA, part number 1SX280LU3F50I2VG. From logic utilization and memory energy consumption (exact numbers are shown in Table C.1 in the Appendix) of convolutional layers we conclude that our method add minor computational overhead in contrast to significant reduction in memory energy consumption. Fig. 5 shows total energy consumption for a single inference of ResNet-18 on ImageNet. In particular, our method is more efficient for higher accuracies, where the redundancy of features is inevitably higher. We further noticed that our approach reached real-time computational inference speed (over 40 fps). The reduction in memory bandwidth can be exploited by using cheaper, slower memory operating at lower clock speeds, which may further reduce its energy footprint. More details appears in the Appendix. Source files for hardware implementation can found at reference implementation.

# 6 Conclusion

This paper presents a proof-of-concept of energy optimization in NN inference hardware by lossy compression of activations prior to their offloading to the external memory. Our method uses transform-domain coding, exploiting the correlations between the activation values to improve their compressibility, reducing bandwidth by approximately 25% relative to VLC and approximately 40% relative to an 8-bit baseline without accuracy degradation and by 60% relative to an 8-bit baseline with less than 2% accuracy degradation. The computational overhead required for additional linear transformation is relatively small and the proposed method can be easily applied on top of any existing quantization method.

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

# A    Additional experimental results

In Fig. A.1 we show Rate-distortion curves on ResNet50, ResNet101 and InceptionV3 for weights quantized to 4 and 8 bits. We show the theoretical entropy value and the real value using Huffman VLC. In Table A.1 we show the comparison of our method with other post-training quantization method. For fair comparison, we add to all compared methods a VLC and show the minimum amount of information that is required to be transferred to the memory.

Table A.1: Comparison of our method against three known post-training quantization methods ((i) ACIQ (Banner et al., 2018); (ii) GEMMLOWP (Jacob & Warden, 2017); (iii) KLD (Migacz, 2017); . We report the smallest bit per value for which degradation is at most 0.1% of the baseline.

| Architecture | Weights (bits) | Method | Activations (avg number of bits per value) |
|---|---|---|---|
| ResNet-50 | 8 | GEMMLOWP | 6.88 |
| | | ACIQ | 6 |
| | | KLD | 6.3 |
| | | Our method | **4.15** |
| | 4 | GEMMLOWP | 6.93 |
| | | ACIQ | 6.2 |
| | | KLD | 6.52 |
| | | Our method | **4.25** |
| Inception V3 | 8 | GEMMLOWP | 6.93 |
| | | ACIQ | 6.2 |
| | | KLD | 6.43 |
| | | Our method | **4.3** |
| | 4 | GEMMLOWP | 6.97 |
| | | ACIQ | 6.3 |
| | | KLD | 6.35 |
| | | Our method | **4.6** |
| MobileNetV2 | 8 | GEMMLOWP | 8.6 |
| | | ACIQ | 7.5 |
| | | KLD | 7.8 |
| | | Our method | **3.8** |
| | 4 | GEMMLOWP | 8.8 |
| | | ACIQ | 7.85 |
| | | KLD | 8.1 |
| | | Our method | **4** |

**Layerwise performance.**    In order to understand the effect of the layer depth on its activation compressibility, we applied the proposed transform-domain compression to each layer individually. Fig. A.2 on the following page reports the obtained rates on ResNet-18 and 50. High coding gain is most noticeable in the first layer, which is traditionally more difficult to compress, and is consequently quantized to higher precision (Hubara et al., 2018; Banner et al., 2018; Choi et al., 2018b).

## A.1    CIFAR-10 results

# B    Block shape and size

Fig. B.1 shows the rate-distortion curves for blocks of the same size allocated differently to each of the three dimensions; the distortion is evaluated both in terms of the MSE and the network classification accuracy. The figure demonstrates that optimal performance for high accuracy is achieved with

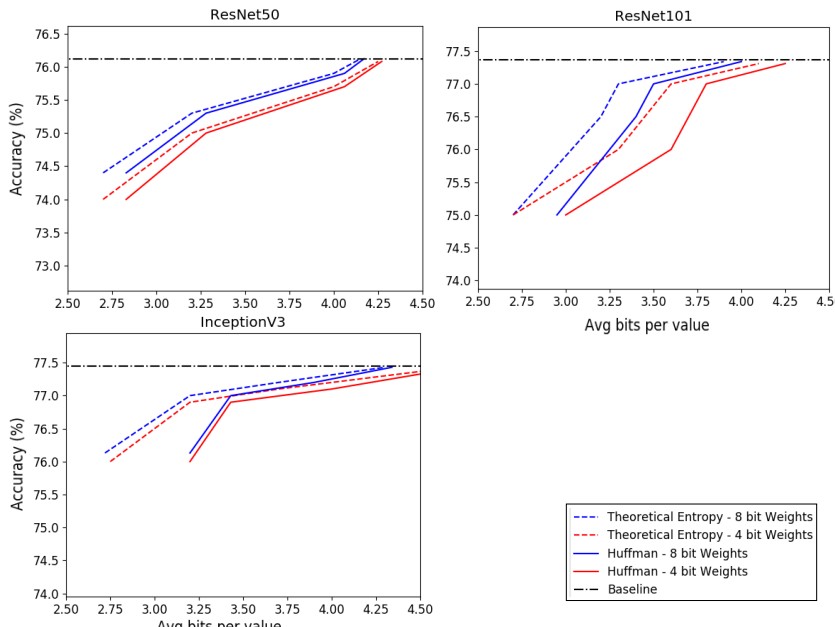

Figure A.1: Rate-distortion curves for ResNet50, ResNet101 and Inception V3 architectures with 8- (blue) and 4-bit (red) weight quantization. Distortion is evaluated in terms of top-1 accuracy on ImageNet. Dashed lines represent rates obtained by Huffman VLC, while solid lines represent the theoretical rates (entropy).

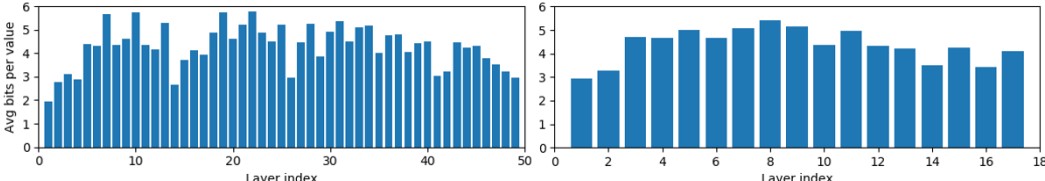

Figure A.2: Theoretic average rates for individual layers in ResNet-50 (left) and Resnet-18 (right); lower values indicate better compression. Both configurations achieve the full precision baseline top-1 accuracy on ImageNet. The first layer is assigned the lowest bitwidth in both models where we observe high correlations between channels.

$1 \times 1 \times C = n$ blocks, suggesting that the correlation between the feature maps is higher than that between spatially adjacent activations. For lower accuracy, bigger blocks are even more efficient, but the overhead of 4 times bigger block is too high. Experiments reported later in the paper set the block size to values between 64 to 512 samples.

## C  Hardware implementation details

We have implemented ResNet-18 using a Stratix-10 FPGA by Intel, part number 1SX280LU3F50I2VG. The memory used for energy calculation is the Micron 4Gb x16 - MT41J256M16. Current consumption of the DDR was taken from the data sheet for read and write operation, and was used to calculate the energy required to transfer the feature maps in each layer. The script for calculating the energy consumption accompanies reference implementation.

In our design each convolutional layer of ResNet-18 is implemented separately. In each layer we calculate 1 pixel of 1 output feature each clock. For example, the second layer has 64 input and 64 output feature maps, thus it takes $64 \times (56 \times 56)$ clock cycles to calculate the output before moving to the next layer.

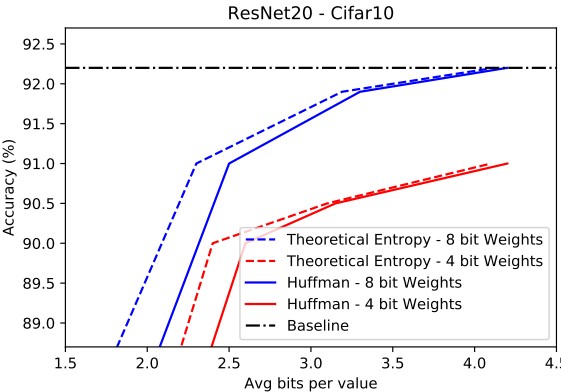

Figure A.3: Rate-distortion curves for ResNet-20 with 8- (blue) and 4-bit (red) weight quantization. Distortion is evaluated in terms of top-1 accuracy on CIFAR-10. Dashed lines represent rates obtained by Huffman VLC, while solid lines represent the theoretical rates (entropy).

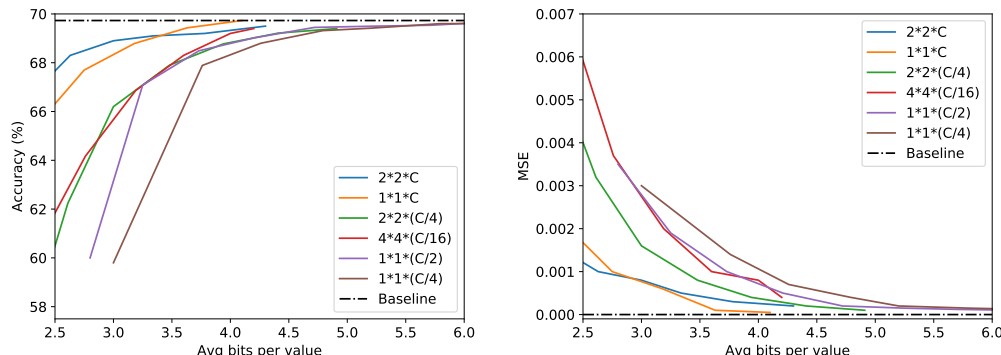

Figure B.1: The influence of the block shape on the top-1 validation accuracy (left) and MSE (right) of ResNet-18 on ImageNet. Our experiments show that for the same size, the most efficient shape is $1 \times 1 \times C$, taking advantage of the correlations across different feature maps at the same spatial location. Theoretical rates are shown.

We read the input features only once by caching the pixels and reusing them from internal memory, and only reloading the weights of the filters in the current layer.

# D  Optimal rate derivation

Let us assume Gaussian activations, for which there are emperical evidence (Lin et al., 2016; Cai et al., 2017; Banner et al., 2018), $x_i \in \mathcal{N}(\mu_i, \sigma_i^2)$, and let $R_i$ bits be allocated to each $x_i$. The optimal $\ell_2$ distortion of $x_i$ achieved by a uniform quantizer constrained by the rate $R_i$ can be approximated as $D_i \approx \frac{\pi e}{6}\sigma_i^2 2^{-2R_i}$. The approximation is accurate until about $R_i \approx 1$ bit (Goyal, 2000). Unfortunately, no general closed-form expression exits relating $R_i$ with the corresponding quantizer step $\Delta_i$, thus, the latter is computed numerically. From our experiments, we derived the approximation $\Delta_i \approx 4.2184 \cdot 2^{-R_i}$, accurate for $R_i \geq 2$ bits.

Optimal rate allocation consists of minimizing the total distortion given a target average rate $R$ for the entire activation block. Under the previous assumptions, this can be expressed as the constrained optimization problem

$$\min_{R_1,...,R_n} \sum_{i=1}^{n} \frac{\pi e}{6}\sigma_i^2 2^{-2R_i} \quad \text{s.t.} \quad \frac{1}{n}(R_1 + \cdots + R_n) = R, \tag{D.1}$$

Table C.1: Logic utilization and memory energy consumption of layers of various widths on Intel's Stratix10 FPGA. Clock frequency was fixed at 160MHz for each design. In LUTs and DSP we present the % of total resources. In Power and Bandwidth we present the total number (% saving comparison to regular quantization)

| # channels | Method | LUTs | DSPs | Energy ($\mu$J) | Bandwidth (Gbps) |
|---|---|---|---|---|---|
| 64 | Quantization | 19K | 960 | 225.93 | 1.28 |
| | Q+VLC | 19K | 960 | 173.44 (-23%) | 0.96 (-25%) |
| | Q+VLC+PCA | 19.5K (+5%) | 1056(+10%) | 100.6 (-44%) | 0.68 (-46.8%) |
| 128 | Quantization | 43K | 2240 | 112.96 | 1.28 |
| | Q+VLC | 43K | 2240 | 148 (-17%) | 1.04 (-18.7%) |
| | Q+VLC+PCA | 45K(+4.3%) | 2366(+5.6%) | 117 (-35%) | 0.83 (-35.1%) |
| 256 | Quantization | 91K | 4800 | 56.5 | 1.28 |
| | Q+VLC | 91K | 4800 | 46.8 (-17.2%) | 1.03 (-19.5%) |
| | Q+VLC+PCA | 93K(+4%) | 5059(+5.4%) | 103 (-42.8%) | 0.73 (-42.9%) |
| 512 | Quantization | 182K | 9600 | 28.2 | 1.28 |
| | Q+VLC | 182K | 9600 | 23.9 (-15.6%) | 1.05 (-17.9%) |
| | Q+VLC+PCA | 186K(+2%) | 10051(+4.7%) | 91 (-49.5%) | 0.66 (-48.4%) |

which admits the closed-form solution (Goyal, 2001), $R_i^* = R + \log_2 \sigma_i - \frac{1}{n}\sum_{k=1}^{n} \log_2 \sigma_k$, and $R_i^* = 0$ whenever the latter expression is negative. The corresponding minimum distortion is given by

$$D^*(R) = \frac{\pi e}{6}\left(\sigma_1^2 \cdot \cdots \cdot \sigma_n^2\right)^{\frac{1}{n}} 2^{-2R}. \tag{D.2}$$

# E    Projection into principal components

In Fig. E.1 we show an intuition of the effect of the projection into the principal components, in 2 layers of ResNet18. This projection helps to concentrate most of the data in part of the channels, and then have the ability to write less data that layers.

# F    Huffman encoder as VLC

Huffman encoder is a known algorithm for lossless data compression. The ability of compress, i.e achieve the theoretical bound of entropy, can be seen in the balance of the huffman tree, means that if the huffman tree is more unbalanced we get better compression. In Fig. F.1 there is an example of huffman tree with and without projection on the principal components, after the projection the huffman tree is more unbalanced.

# G    Eigenvalues analysis

The eigenvalues of the covariance matrix is a measure of the dispersal of the data. If high energy ratio, means the cumulative sum of eigenvalues divided by the total sum, can be expressed with small part of the eigenvalues, the data is less dispersal and therefore more compressible. In figure G.1 we analyze the covariance energy average ratio in all layers of different architectures. The interesting conclusion is that the ability of compression with the suggested algorithm is correlated with the covariance energy average ratio , means that for new architectures we can look only at the energy ratio of the activation to measure our ability of compression.

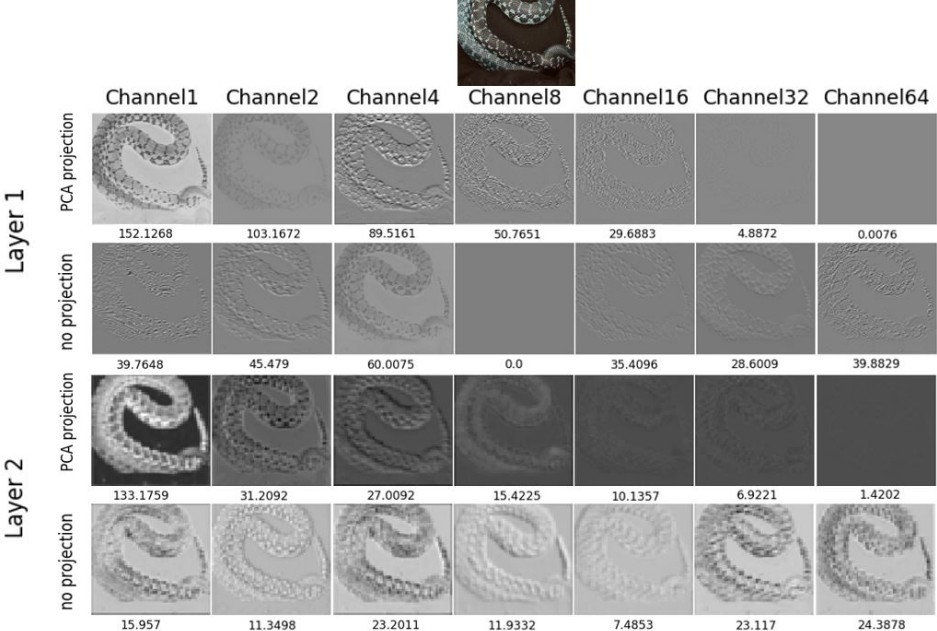

Figure E.1: Channels from layers one and two in ResNet18 with and without principal component projection. Using PCA we create new channels where each channel is based on all original channels, ordered by how well the new channels capture the data. Under each channel we can see the energy (in terms of L2 norm) associated with that channel. Most energy is now concentrated within a few known channels (the first channels), enabling aggressive quantization for the rest of the channels. This is unlike the case of no projection where energy is more spread out among the different channels without affording an easy way to prioritize between them. Finally, note that values of the last channels are almost identical. For example, all values of channel 64 are mapped to a single value $v$, which can benefit the entropy encoder by using a short codeword for $v$ and thus write less bits to memory for that channel.

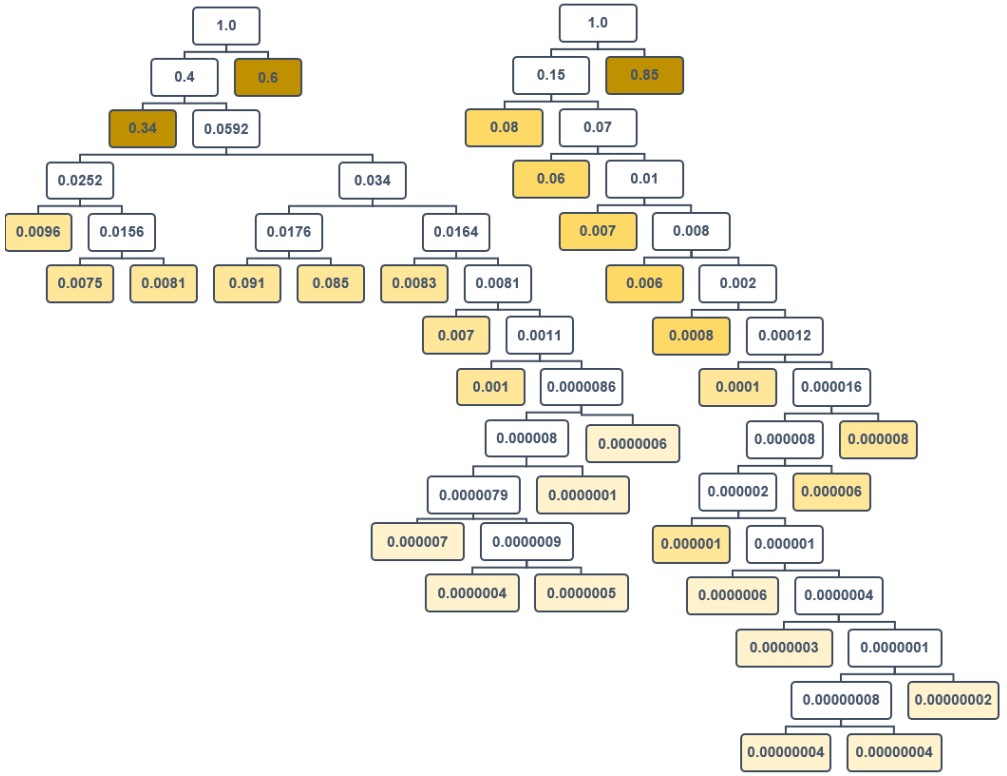

Figure F.1: Example of Huffman tree with (Right) and without (Left) projection on the principal components. The colored nodes represent the leaves of the tree. We can see that when we project on the principal component the tree is less balanced, means can be better compressed

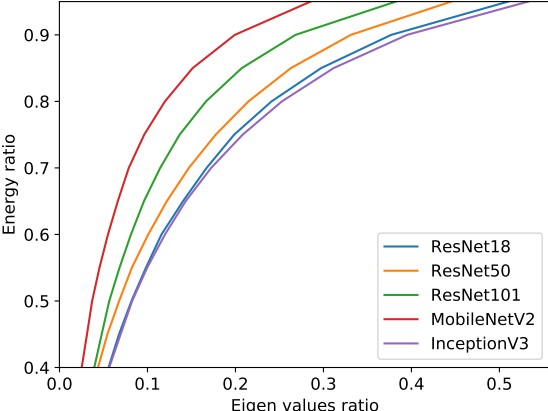

Figure G.1: Analysis of the eigenvalues ratio that are needed to achieve energy ratio, means cumulative sum of the eigenvalues.

