# OpenReview forum: "Feature Map Transform Coding for Energy-Efficient CNN Inference"
_ICLR.cc/2020/Conference — Reject_

### Official Review · AnonReviewer2 · 2019-10-22
**Official Blind Review #2**

**Rating:** 3

**Review:**

A lossy transform coding approach was proposed to reduce the memory bandwidth of edge devices deploying CNNs. For this purpose, the proposed method compresses highly correlated feature maps using variable length coding. In the experimental analyses, the proposed method outperforms some of the previous work in terms of the compression ratio and accuracy for training ResNet-34 and MobileNetV2.

The proposed method and initial results are promising. However, the paper and the work should be improved for a clear acceptance:

- Some parts of the method need to be explained more clearly:

– In the statement “Due to the choice of 1 × 1 × C blocks, the PCA transform essentially becomes a 1 × 1 tensor convolution kernel”, what do you mean by “the PCA transform becomes a convolution kernel.”?

- Could you please further explain how you compute PCA using batch data, how you update online and how you employ that in convolution weights together with BN? Please also explain the following in detail:

(I) “To avoid this, we calculate the covariance matrix layer by layer, gradually applying the quantization.” What is the quantization method you applied, and how did you apply it gradually?

(II) “The PCA matrix is calculated after quantization of the weights is performed, and is itself quantized to 8 bits.” How did you quantize the weights, how did you calculate PCA using quantized weights and how did you quantize them to 8 bits?

- Could you please explain the following settings, more precisely: direct quantization of the activations; quantization of PCA coefficients; direct quantization followed by VLC; and full encoder chain comprising PCA, quantization, and VLC? Please note that there are various methods and algorithms which can be used for these quantization steps. Therefore, please explain your proposed or employed quantization methods more clearly and precisely.

–  Please clarify the statement “This projection helps to concentrate most of the data in part of the channels, and then have the ability to write less data that layers.”.

- Did you apply your methods to larger networks such as larger ResNets, VGG like architectures, Inception etc?

- I also suggest you to perform experiments on different smaller and larger datasets, such as Cifar 10/100, face recognition datasets etc., to examine generalization of the proposed methods at least among different datasets.

**Experience Assessment:**

I have published in this field for several years.

**Review Assessment: Checking Correctness Of Derivations And Theory:**

I carefully checked the derivations and theory.

**Review Assessment: Checking Correctness Of Experiments:**

I carefully checked the experiments.

**Review Assessment: Thoroughness In Paper Reading:**

I read the paper at least twice and used my best judgement in assessing the paper.

---

> ### Author Response · Authors · 2019-11-10
> **Answer to Reviewer #2**
>
> Thank you very much for your review, following are the answer to your concerns:
>
> 1. In every 3-dimensional tensor (feature map), the PCA transform can be applied to various block shapes. In Fig B.1 we checked it and the more efficient shape was 1 x 1 x C. Choosing this shape has a big implementation advantage because it can be implemented using the convolution kernel (which is very efficient), with kernel size 1 x 1 and where the weights (along the C channels) are exactly the principal components.
>
> 2. The work is in post-training regime means there is no labeled data and we do not run backpropagation. The PCA is calculated only on  a single batch (calibration). After we calculate it, the PCA is fixed (as the convolutional weights) and is not changed - in that way it is much more efficient since the calculation of the PCA matrix is computationally expensive.
>
> About your question of employing the convolution together with BN (known as “folding”): this is a common technique employed in hardware to reduce the amount of computation, described, for example in “Quantization and Training of Neural Networks for Efficient Integer-Arithmetic-Only Inference” by Jacob et al. In the same way, we fold the PCA into the previous convolutional layer for saving arithmetic complexity. We added a reference to the above mentioned paper.
>
> 3. “To avoid this, we calculate the covariance matrix layer by layer, gradually applying the quantization.” - We apply uniform quantization to all layers of the network. The idea of gradual quantization for the covariance matrix means that we first quantize the first layer and calculate its covariance matrix; only after quantizing the first layer, we proceed with quantizing the following layer (instead of quantizing all layers at the same time) - The idea behind this is that the covariance matrix includes the real statistics of the network that is affected by the quantization of the previous layers.
>
> 4. “The PCA matrix is calculated after quantization of the weights is performed, and is itself quantized to 8 bits.”  - The weights and the PCA coefficients are quantized to 8 bits with standard uniform quantization (specifically, a mid-tread uniform quantizer to ensure 0 is one of the bins). The PCA matrix of the feature map k is calculated after the weights of convolution k are quantized to 8 bits, so the PCA contain the real statistics of the activations produced at inference.
> 5. In Figure 4 we show the efficiency of each part of the suggested algorithm:
> “direct quantization of the activations”:
> * Only quantization of the feature maps with standard uniform quantization - marked as “Q” in Figure 4 left.
> * quantization of PCA coefficients - applying PCA transform to the feature maps and quantizing the latter - marked as PCA —> Q in Figure 4 left.
> * direct quantization followed by VLC - Applying quantization to the feature maps and then compressing them using VLC (No PCA) - marked as Q —> VLC in Figure 4 left.
> * full encoder chain comprising PCA, quantization, and VLC - The full suggested method, including: applying PCA, quantizing the coefficients, and then applying VLC - marked as PCA —> Q —> VLC in figure 4 left.
> The figure suggests that the full method achieves highest performance.
>
> 6. “This projection helps to concentrate most of the data in part of the channels, and then have the ability to write less data that layers.” - In figure E.1 we show what happens to the image after the projection onto the principal components. Because of the high correlation between channels we can see that after the projection more information is concentrated in the first channels, while the last channels are almost constant. This shall be compared to the case where there is no projection and the information is spread across all channels. Concentration of information in a small number of coefficients is the key tool for achieving the high compressibility reported in the paper.
>
> 7. Results of Inception V3 and other methods are reported in the appendix, Figure A.1 and Table A1.
>
> 8. Following your suggestion, we will add to the new version results for a smaller dataset (CIFAR10, Figure A.3 in the appendix). We are also checking the generalization of the proposed method to other tasks.

---

### Official Review · AnonReviewer3 · 2019-10-24
**Official Blind Review #3**

**Rating:** 8

**Review:**

The submission proposes to reduce the memory bandwidth (and energy consumption) in CNNs by applying PCA transforms on feature vectors at all spatial locations followed by uniform quantization and variable-length coding.

I appreciate the writing quality: as an outsider to the field of low-power/low-precision deep learning, I found the write-up straightforward and easy to follow. It’s harder for me to precisely assess the significance of the proposed approach, but at a high level it looks reasonable and is backed by convincing empirical evidence.

Small comment: I don’t believe the submission is following the ICLR 2020 format strictly: the font looks different, and the margins look tighter.

**Experience Assessment:**

I do not know much about this area.

**Review Assessment: Checking Correctness Of Derivations And Theory:**

N/A

**Review Assessment: Checking Correctness Of Experiments:**

I assessed the sensibility of the experiments.

**Review Assessment: Thoroughness In Paper Reading:**

N/A

---

> ### Author Response · Authors · 2019-11-10
> **Answer to Reviewer #3**
>
> Thank you very much for your comments and rating. As proposed, we uploaded a fixed version. For some reason one of the TeX packages interfered with it - we apologize for that.

---

### Official Review · AnonReviewer4 · 2019-11-05
**Official Blind Review #4**

**Rating:** 3

**Review:**

This paper studies an important question: how to reduce memory bandwidth requirement in neural network computation and hence reduce the energy footprint. It proposes to use lossy transform coding before sending network output to memory. My concern with the paper is two-fold:
1) The major technique of transform-domain coding is borrowed from previous work (e.g., Goyal 2001), hence the novelty of the proposed method is in doubt.
2) The implementation details are not clear. For example, I don't know whether the implementation in section 3.1 is based on CPU or FPGA, and how easily Section 3.1 will be implemented on ASIC. For the experimental results are reported in Section 4, we do not know how much memory and how much cache is used. Will the computation of PCA require a lot of on-device memory?

More detailed comments:
Section 1, 2nd paragraph: GPUs are event more popular than FPGAs and ASICs. Can the proposed method be useful for GPU inference?
Section 1, 3nd paragraph:  The last sentence says "high interdependence between the feature maps and spatial locations of the compute activations". However, it is not clear to me how the proposed method takes spatial location into account.
Section 2: better to review previous work In lossy transform coding
Figure 1: It seems to me Figure 1 is obvious. What is the novelty?
Section 4: better to report the details of computing units and memory size.

**Experience Assessment:**

I have read many papers in this area.

**Review Assessment: Checking Correctness Of Derivations And Theory:**

I assessed the sensibility of the derivations and theory.

**Review Assessment: Checking Correctness Of Experiments:**

I assessed the sensibility of the experiments.

**Review Assessment: Thoroughness In Paper Reading:**

I read the paper at least twice and used my best judgement in assessing the paper.

---

> ### Author Response · Authors · 2019-11-10
> **Answer to Reviewer #4**
>
> We thank the reviewer for the detailed comments. In what follows, we address in detail the raised issues.
>
> 1. The transform coding theory is based on previous work — indeed, we referred to (Goyal 2001) as well as much older works in the field of image and video compression. However, its use in for neural networks showing the correlation that can be exploited to reduce the memory bandwidth in the activations tensors is novel and was not shown before. In addition, we showed a reference hardware implementation that confirms this theory.
>
> 2. The implementation is divided into 2 parts:
> A PyTorch implementation of the algorithm, including various modern architectures.
> A reference implementation on an Altera FPGA that confirms the reduction in memory energy consumption during inference
> Both parts are fully replicable using the code that accompanies the paper. ASIC mplementation should be straightforward using the provided RTL —  we chose the FPGA target due to the easier prototyping cycle.
>
> Regarding the use of cache and memory: in this work, we focus on compression of the feature maps, since in modern systems the cache is insufficiently big to contain all the feature maps; for this reason, in every forward path, writes to the external DDR are inevitable. It was shown in (Yang et al., 2017) that this data movement is a significant constituent of the energy footprint. In our FPGA implementation, we used small buffers and no associative cache memories on the path to/from the DDR.
>
> The computation of PCA does not require a lot of on-chip memory. In fact, it can be interpreted as another 1x1 convolution. It adds a certain computational overhead as detailed in Table C.1; yet, because of the efficient implementation of the convolution, it is negligible in comparison to the benefit in bandwidth reduction.
>
> 3. The method can be used in any system where memory bandwidth significantly contributed to the energy footprint. This includes GPU-based systems. However, in order to be efficient, it requires hardware acceleration of certain operations such as VLC/VLD in the memory hierarchy, which currently lacks in existing GPUs.
>
> 4. The method exploits spatial dependencies of the activations by coding blocks from the activation tensor. Figure B.1 visualizes the amount of compression achieved by different block configurations across the activation channels and spatial dimensions. The highest correlation was found across the different channels at the same spatial location.
>
> 5. Table C.1 contains more details about the logic utilization and the memory energy consumption in the hardware implementation.

---

### Decision · Program_Chairs · 2019-12-19

**Decision:**

Reject

**Comment:**

The paper proposed the use of a lossy transform coding approach to to reduce the memory bandwidth brought by the storage of intermediate activations. It has shown the proposed method can bring good memory usage while maintaining the the accuracy.
The main concern on this paper is the limited novelty. The lossy transform coding is borrowed from other domains and only the use of it on CNN intermediate activation is new, which seems insufficient.